# Optimizing Sparse Testing for Genomic Prediction of Plant Breeding Crops

**DOI:** 10.3390/genes14040927

**Published:** 2023-04-17

**Authors:** Osval A. Montesinos-López, Carolina Saint Pierre, Salvador A. Gezan, Alison R. Bentley, Brandon A. Mosqueda-González, Abelardo Montesinos-López, Fred van Eeuwijk, Yoseph Beyene, Manje Gowda, Keith Gardner, Guillermo S. Gerard, Leonardo Crespo-Herrera, José Crossa

**Affiliations:** 1Facultad de Telemática, Universidad de Colima, Colima 28040, Mexico; 2International Maize and Wheat Improvement Center (CIMMYT), Km 45, Carretera Mexico-Veracruz, El Batan, Texcoco 56237, Mexico; 3VSN International, Hemel Hempstead HP2 4TP, UK; 4Centro de Investigación en Computación (CIC), Instituto Politécnico Nacional (IPN), Mexico City 07738, Mexico; 5Centro Universitario de Ciencias Exactas e Ingenierías (CUCEI), Universidad de Guadalajara, Guadalajara 44430, Mexico; 6Department of Plant Science Mathematical and Statistical Methods—Biometrics, P.O. Box 16, 6700AA Wageningen, The Netherlands; 7Colegio de Postgraduados, Montecillos 56230, Mexico

**Keywords:** sparse testing, wheat, maize, genomic prediction, multi-trait, and uni-trait

## Abstract

While sparse testing methods have been proposed by researchers to improve the efficiency of genomic selection (GS) in breeding programs, there are several factors that can hinder this. In this research, we evaluated four methods (M1–M4) for sparse testing allocation of lines to environments under multi-environmental trails for genomic prediction of unobserved lines. The sparse testing methods described in this study are applied in a two-stage analysis to build the genomic training and testing sets in a strategy that allows each location or environment to evaluate only a subset of all genotypes rather than all of them. To ensure a valid implementation, the sparse testing methods presented here require BLUEs (or BLUPs) of the lines to be computed at the first stage using an appropriate experimental design and statistical analyses in each location (or environment). The evaluation of the four cultivar allocation methods to environments of the second stage was done with four data sets (two large and two small) under a multi-trait and uni-trait framework. We found that the multi-trait model produced better genomic prediction (GP) accuracy than the uni-trait model and that methods M3 and M4 were slightly better than methods M1 and M2 for the allocation of lines to environments. Some of the most important findings, however, were that even under a scenario where we used a training-testing relation of 15–85%, the prediction accuracy of the four methods barely decreased. This indicates that genomic sparse testing methods for data sets under these scenarios can save considerable operational and financial resources with only a small loss in precision, which can be shown in our cost-benefit analysis.

## 1. Introduction

To meet the demands of the growing global population, food production must increase, which is challenging because of the drastic fluctuations in climatic conditions, competition for land and deterioration of natural resources. For this reason, we must adopt novel alternatives for genetic improvement that can increase yield productivity, yield stability, improve disease resistance, nutrition, and the subsequent end-use quality of key crops such as maize, wheat and rice [1].

In this vein, the genomic selection (GS) methodology uses statistical machine learning algorithms and data focused on genomic information to improve the selection of candidate lines, which is key for making crop breeding processes more efficient. GS is a predictive methodology proposed by Meuwissen [2] that trains a statistical machine learning method with data containing phenotypic and genotypic information. This trained model then predicts breeding values or phenotypic values of new (untested) lines that were only genotyped, meaning that lines can be selected earlier [3]. Successful GS methodology is found in many crops such as wheat, maize, cassava, rice, chickpea, groundnut, etc. [4,5,6]. However, the practical implementation of GS for breeders across the world is challenging because GS methodology and genomic prediction (GP) accuracy do not always guarantee moderate to high accuracies, as there are many other factors affecting prediction performance.

To increase genetic gain, breeders must accurately predict breeding values. This is easy when the traits of interest have a simple genetic architecture; however, this is more challenging in traits such as grain yield with a complex, difficult-to-understand genetic architecture. For example, in complex trait prediction, it is difficult to accurately model genetic interactions such as epistatic effects, which are common in plant and animal sciences, as well as in biology [7,8,9,10]. For this reason, some model strategies are more appropriate to capture these complex interaction effects more efficiently, quantifying the level of influence in understanding the genetic architecture of these traits.

Part of the challenge in plant breeding is selecting candidate genotypes that work both across and for specific environmental conditions. Genotypes were evaluated in multi-environmental trials (METs), where the goal is to select stable genotypes across environments and in specific environments considering the genotype × environment (GE) interaction. Precisely evaluating all genotypes once in each environment (that is, each environment is a complete replica of all lines) is more expensive, as it requires more extensive field-testing evaluations [11,12].

“Sparse testing” in GS means that some lines have been evaluated in some environments and only predicted (not observed) in others. For this reason, sparse testing can save resources and can help improve the efficiency of the GS methodology by (a) increasing the number of lines tested and maintaining a fixed number of environments and financial costs or (b) increasing the number of testing environments while maintaining the cost and a fixed number of lines under evaluation. Sparse testing in plant breeding and genome prediction implies modifying the original multi-environment breeding trial system into a testing method where not all lines are sown in all environments because costs and availability of seed, land and water might impede observing all genotypes in all environments. The fundamental question is how to establish a multi-environmental trial system that will be economically acceptable without affecting the precision with which the performance of breeding lines is assessed, predicted, and selected. 

The information provided by the molecular markers assists breeders in predicting unobserved lines in some environments. Although, in most cases, it is impossible to evaluate all lines in all environments, observing some of these lines offers the possibility of assessing the marker alleles (or haplotypes) in all environments and the marker (or haplotype) × environment interaction. Therefore, the information on the response patterns of the markers can be used to improve the predictive ability of the unobserved lines in the environments. By using genome-enabled prediction when modeling genotype × environment, the unobserved genotype × environment combinations can be better predicted, and thus the overall costs of the testing can be reduced.

Recently, Jarquín [13] and Crespo [1] studied genomic sparse evaluation in the context of maize and wheat genomic prediction, including extreme cases of (a) non-overlapping lines between environments, all lines tested in different environments; (b) lines completely overlapping across environments, all lines field evaluated in all the environments; and (c) varying numbers of different overlapping/non-overlapping lines. The results obtained by Jarquín [13] in maize and Crespo [1] in wheat multi-environment trials showed that the genome-based model, including genomic × environment interaction (GE), captured more portions of the total phenotypic variation than the models that did not include this component and provided higher prediction accuracy than other genomic prediction models that did not include GE when applied to multiple sparse testing designs. Thus, both studies clearly show that using sparse testing based on overlapping/non-overlapping methods can lead to substantial savings in testing resources when using appropriate GE genome-based models. However, the methods of Jarquín [13] in maize and Crespo [14] in wheat for assigning lines to environments by the overlapping/non-overlapping were based on a random assignation of lines to environments without any allocation optimization criterion. Also, it should be noted that these studies performed only uni-trait prediction. 

This study aimed to optimize allocation methods to improve the genomic prediction accuracy of sparse testing by evaluating four genomic sparse testing strategies for allocating cultivars to environments. This study addressed four objectives that have not been investigated in any previous studies: (1) to determine if there are differences in prediction ability between the four genomic sparse testing allocation methods; (2) to study if there are significant differences between the four strategies of sparse testing under a uni-trait (UT) and multi-trait prediction framework; (3) to evaluate the performance of the four sparse testing strategies with large and small trials; and (4) to quantify the various benefits of implementing this genomic sparse testing allocation of lines to environments strategies. To achieve these objectives, two real data sets from CIMMYT were used—one maize and one wheat—with one data set containing over 450 lines and the other over 4500 lines. To assess performance with small trial sizes from each of these two data sets, a random sample of 250 lines in each environment was obtained, and the four sparse testing methods were evaluated using this resulting data set.

## 2. Material and Methods

### 2.1. Data Sets

#### 2.1.1. Wheat Data

The original data set contains 4536 lines evaluated in four environments (B2IR, B5IR, BDRT, BLHT). The experimental design employed for arranging all the lines in each environment was an augmented row-column design [14,15] established using the DiGGeR package [16]. Due to some missing plots, only 4464 lines were ultimately evaluated in four environments (B2IR, B5IR, BDRT, BLHT). Four traits were evaluated: grain yield (ton per hectare), days to germination (Germination), days to heading (Heading) and plant height (cm). Since all lines were evaluated in each environment, the total number of observations in this data set is 17,856. This data set was used for the univariate implementation of the models but presented a high unbalance for the multi-trait implementation; hence, we implemented the multi-trait model with a subset of this original data set and guaranteed the presence of the response variables (traits) of all lines and environments. This subset contains 4437 lines, three environments (B2IR, B5IR, BDRT) and the same four traits. 

We performed a mixed model spatial analysis for grain yield in each environment and thus adjusted the data for local and overall spatial variability by spatial adjustment (autoregressive in the directions of rows and columns, AR1 × AR1) using ASReml-R [17]. The weighted BLUEs for each location were used to implement the prediction model described in the next statistical model section. When the complete data were used, this was denominated as the big wheat data set, but when only the sample of 250 lines was, this was called the small wheat data set.

#### 2.1.2. Maize Data

This data set contains 484 lines evaluated in locations within three major environments: drought stress environment (WS), low nitrogen environment (LN) and well water environment (WW). The traits evaluated were grain yield and plant height. Since all lines were evaluated in each environment, the total number of observations was 1452. The experimental design at each location for each major environment (WW, WS, and LN) was an α-lattice design with two replications.

For the maize data, we used the two-stage analysis to initially account for the within-environmental variance in the first stage and to assess the genomic and genomic × environment effect in the second stage. The first-stage analysis consisted in computing the best linear unbiased estimates (BLUEs) of the maize testcrosses across locations for each major environment (WW, WS, LN) using the following linear mixed model:$$Y_{\mathrm{ijk}}{=\mu +R}_{r}{+B}_{k}\left[R_{r}\right]{+G}_{i}{+\varepsilon }_{\mathrm{ijr}}$$
where $Y_{\mathrm{irk}}$ is the response variable of testcross *i* at replicate *r* within the incomplete block *k*; *µ* is the general mean; $R_{r}$ is the fixed effect of the replicate *r*; $B_{k}\left[R_{r}\right]$ is the random effect of the incomplete block *k* within replicate *r* assumed to be independently and identically normally distributed with mean zero and variance $\sigma_{B(R)}^{2}$; $G_{i}$ is the fixed effect of genotype *i*; and $\varepsilon_{\mathrm{irk}}$ is the random residual error assumed independent and identically normally distributed with mean zero and variance $\sigma_{\varepsilon }^{2}$.

This weighted BLUE data set in each major environment WW, WS, and LN was used for the evaluation of the uni-trait and multi-trait methods. From this data, a smaller data set with 250 lines was created to evaluate the performance of the four sparse methods under both the uni-trait and multi-trait frameworks. When we aggregate the summaries of the prediction performance of the two big (wheat and maize) and two small (wheat and maize) data sets, we call this across data sets.

### 2.2. Statistical Model

This model was used for the training process of the sparse testing designs: (1)$$Y=1_{n}\mu^{T}+X_{E}\beta_{E}+Z_{L}g+Z_{EL}gE+\epsilon$$
where Y is the matrix of phenotypic response variables of order $n\times n_{T}$, ordered first by environments and then by lines; $n_{T}$ denotes the number of traits, $1_{n\times n_{T}}$ is a matrix of ones of order $n\times n_{T}$, $\mu^{T}$ is a vector of intercepts for each trait of length $n_{T}$, T denotes the transpose of a vector or matrix, that is, $\mu ={\left[\mu_{1},\ldots ,\mu_{n_{T}}\right]}^{T},X_{E}$ is the design matrix of environments of order n×I, I denotes the number of environments, $\beta_{E}$ is the matrix of beta coefficients for environments with a dimension of $I\times n_{T}$, $Z_{L}$ is the design matrix of lines of order n×J, J denotes the number of lines, g is the matrix of random effects of lines of order $J\times n_{T}$ distributed as $g∼MN_{J\times n_{T}}\left(0,G,\Sigma_{T}\right)$, that is, with a matrix-variate normal distribution with parameters M=0, U=G and $V=\Sigma_{T}$, G is the genomic relationship matrix [18] built with marker data of order J×J and $\Sigma_{T}$ is the variance-covariance matrix of traits of order $n_{T}\times n_{T}$. $Z_{EL}$ is the design matrix of the genotype × environment interaction of order n×JI, gE is the matrix of genotype × environment interaction random effects distributed as $gE∼MN_{JI\times n_{T}}\left(0,Z_{E}Z_{E}^{T}⨀Z_{g}GZ_{g}^{T},\Sigma_{T_{2}}\right)$, where $\Sigma_{T_{2}}$ is the variance-covariance matrix of traits of order $n_{T}\times n_{T},\;⨀$ denotes the Hadamard product. ϵ is the residual matrix of dimension $n\times n_{T}$ distributed as $\epsilon ∼MN_{n\times n_{T}}\left(0,I_{IJ},R\right)$, where R is the residual variance-covariance matrix of order $n_{T}\times n_{T}$. This model was conducted in the BGLR library [19]. Moreover, a uni-trait version of this model given in equation (1) was implemented, assuming that the response variable (Y) was a vector, that is, training the model with only one trait at a time. 

### 2.3. Sparse Testing Methods for the Allocation of Lines to Environments 

We used the notation J as the number of lines, k as the number of lines per location, I as the number of environments (locations) and r as the number of replications for the *j*th line in the entire design. It should be noted that since the four methods are based on the incomplete block principle, k is less than J, since not all lines in each environment can be assigned. An equal concurrence of entries by location is the best way to ensure minimum variance when making all pair-wise comparisons. Therefore, since $r_{i}=r$ for all lines, the total number of observations in the experiment is N, where N=J×r=I×k.

#### 2.3.1. Method 1 (M1)-Allocation of Fraction of Lines in All Locations

This was the simplest allocation method where a fraction (subset) of lines is selected and then allocated in all locations as a training set where the remaining lines are used as the testing set. In Figure 1A, we see how the partition is formed with this method; blue represents the lines used as the training set, and white represents the lines used as the testing set. Training lines are grouped at the beginning of Figure 1A, but the lines are not in numerical order, indicating they were randomly selected.

#### 2.3.2. Method 2 (M2)-Allocation of Fraction of Lines with Some Shared Lines in Locations 

M2 took a fraction (subset) of lines to be used as a training set and the remaining as a testing set in one location. For the other locations, the testing lines were divided into a number of locations—one part that is ideally the same size and one part that is interchanged from testing to training for each location. In this way, each location shared most of the training lines but contained some lines only in testing, as shown in Figure 1B.

#### 2.3.3. Method 3 (M3)-Random Allocation of Fraction of Lines to Locations under Incomplete Locations 

Starting from a balanced data set with J lines and I locations, the conformation of the random allocation of lines to locations was done in such a way that each line will be repeated (approximately) in r out of I locations, and all locations will be of the same size (k). The algorithm of this random allocation is [20]:

First, we computed $k=\frac{J\times r}{I}$ (least integer greater than or equal to $k=\frac{J\times r}{I}$). Then k lines out of J lines were randomly allocated to the first location. For the second location, k out of the J lines were once again randomly allocated. This process is repeated until the Ith location is completed, with the caveat that the lines allocated to a particular location are only present in less than or equal to r locations, ideally in exactly r locations. The lines that do not satisfy this restriction are not candidates for being allocated to a particular location.

An example of this method with eight lines and four locations is shown in Figure 1C. Note that some locations appear up to three times in each line, while others appear only twice because it is a random process.

#### 2.3.4. Method 4 (M4)-Allocation of Lines to Locations Using the IBD Principle

This method of allocation of lines to environments is based on a balanced incomplete block design (IBD) principle, that is, when all pairs of lines occur together within a location an equal number of times (λ). In general, we specified $\lambda_{jj}$ as the number of times line j occurs within a location. To generate this sparse allocation of lines to locations [20], we used the function find.BIB() in the R package crossdes. Supposing there were J=8 lines and I=4 locations, this means that we need 8×4=32 plots to allocate the eight lines to the four locations. However, we used an IBD and a training set of size $N_{TRN}=32\times 0.5=16$, which accounts for (50%) of the total plots required under a randomized complete block design. Therefore, the number of lines by locations can be obtained by solving (kI=N_TRN) for k, which results in k=N_TRN/I. This results in k=16/4=4 lines per location. The corresponding elements for the training set were obtained with the function find.BIB(8, 4, 4) using the package crossdes. The numbers used in the function find.BIB() denote the lines, the locations, and the lines per location, respectively. Figure 1D shows how a particular allocation for eight lines in four locations may appear. An important aspect to consider is that both lines are used at the same time, and all locations contain four lines. The final allocations of M3 and M4 in many cases look similar, with the relevant difference that M4 is allocated under the IBD principles, while M3 is under a kind of stratified random sampling. 

Since not all lines (treatment structure) will be allocated to each environment (plot structure), the four allocation methods described are sparse allocation methods. Each of these methods allocates lines to environments (locations) under different approaches, and some of them guarantee better connectivity of lines between environments, and for this reason, they provide different prediction performances. However, as pointed out by Montesinos-López [20], implementing the sparse allocation methods for genomic prediction is done in two stages, so each stage should be optimized. The four methods of allocation (M1 to M4) belong to the second stage, and these four allocation methods attempt to optimize the prediction performance of untested lines in tested environments. 

To obtain valid results in the second stage, we used valid BLUEs or best linear unbiased predictions (BLUPs) for each line. An optimal experimental design should be used in the first stage for allocating the lines that were allocated for each environment with any of the four methods of the second stage, in each environment to plots. Our two-stage approach of analysis is valid since it is like the BLUEs or BLUPs estimation in two stages, and there is strong empirical evidence that two-stage analysis produces similar outputs when the appropriate weighting methods are used [21,22].

### 2.4. Cross-Validation Strategy

To evaluate and compare the predictive performance of the allocation methods, we used cross-validation with 10 partitions and 15, 25, 50, 75 and 85% of the data for training and 85, 75, 50, 25 and 15% for testing, respectively. The Pearson’s correlation and the Normalized Root Mean Squared Error (NRMSE) were computed using the observed and predicted values [23] in each of the 10 random partitions with the testing sets. These metrics were then used to assess the predictive performance in each data set for each allocation method. The average of the NRMSEs and Average Pearson’s correlations (APC) of the 10 partitions was reported as prediction accuracy in each data set. Since the allocation methods were evaluated under uni-trait and multi-trait frameworks, both metrics were computed for each trait separately, and then, the average of the 10 folds in the testing sets was reported as prediction performance. It is important to point out that we used different proportions of the testing set (15, 25, 50, 75 and 85%) to study that even with a small proportion of training sets, it is feasible to predict the testing set with reasonable accuracy.

## 3. Results

The results are provided in four sections; one for each complete (big) data set in maize and wheat, one for the results across data sets (summary of aggregating the four data sets: two big data sets and two small data sets) and one that illustrates the quantitative benefits of sparse testing methods when using all data. Figures and tables for the results obtained for the small data sets of maize and wheat (random selection of only 250 lines for each data set) are provided in the Appendix A for Maize_250 small data set and Appendix A for Wheat_250 small data set).

### 3.1. Complete Maize Data Set (Big Maize Data Set)

In terms of APC, in all scenarios of testing proportions, the best prediction performance was observed under a multi-trait framework and the worst under a uni-trait framework (Figure 2, Table A1). Under the scenario with 85% (0.85) of testing, the GP accuracy does not deteriorate and is only slightly worse than under the scenario of predicting 15% testing. Between the four sparse testing methods, we observed no significant differences between them. Instead, we saw a relevant difference of 15% (0.15), 75% (0.75) and 85% (0.85) of testing M4 under the uni-trait framework. For example, under the 15% (0.15) of testing set uni-trait framework, M4 outperformed M1, M2 and M3 by 12.89, 7.08 and 7.33%, respectively. Under the 85% (0.85) testing uni-trait framework, M4 only outperformed methods M1 and M2 by 1.6% and 4.9%, respectively, but no difference was observed between methods in most of the proportion of testing evaluated (Figure 2, Table A1). 

In terms of NRMSE, we observed the best predictions in a multi-trait model and the worst in a uni-trait model (Figure 3, Table A1). In general, we also observed that the best predictions in terms of NRMSE were under methods M3 and M4. When the percentages of testing were 15% (0.15), 25% (0.25) and 50% (0.5), small differences were found between the four sparse testing methods; however, under 75% (0.75), methods M1 and M2 were worse than M3 by 2.3% and 56.33% (multi-trait) and by 4.6% and 31.6% (uni-trait). Under 85% (0.85), method M4 outperformed methods M1 and M2 by 5% and 60.6% (multi-trait) and by 3.7% and 30.5% (uni-trait) (Figure 3, Table A1).

### 3.2. Complete Wheat Data Set (Big Wheat Data Set)

In this data set in terms of APC, the multi-trait method was better than the uni-trait framework (Figure 4, Table A2). While we did not observe a superiority in all percentages of testing of a particular method, we noted that in all scenarios of percentages of testing, the prediction accuracies are quite similar; that is, the prediction performance does not decrease as the percentage of testing increases. Regarding the comparison between the four sparse methods, we observed that in some scenarios of percentage of testing, M4 was better than the remaining methods. For example, under 15% (0.15) of the uni-trait model, testing set M4 outperformed M1 and M2 by 24.13 and 24.08, respectively, but did not outperform M3, while under the 25% (0.25) uni-trait testing, M4, M1 and M3 outperformed method M2 around 25%. In the remaining cases, no relevant differences were observed between methods in most of the proportion of testing evaluated (Figure 4, Table A2). It is important to note that for some scenarios, we were unable to estimate the prediction performance for methods M3 and M4 due to a lack of positive definite matrices for multi-trait models.

In terms of NRMSE, the best predictions were obtained under a multi-trait method and the worst under a uni-trait method (Figure 5, Table A2). We also observed that under the 15% (0.15) and 25% (0.25) multi-trait framework, M3 was slightly better than the others. While under the 50% (0.50) multi-trait framework, M3 and M4 were slightly better than M1 and M2. In a small number of cases in the uni-trait model, M1 was better than the remaining methods (Figure 5, Table A2).

### 3.3. Across Data Sets

Across data sets, the best predictions were observed under the multi-trait model in terms of APC (Figure 6, Table A3). In most percentages of testing, there were no relevant differences among the four methods. For example, when the percentage of testing was 15% (0.15) in the uni-trait model, methods M3 and M4 outperformed methods M1 and M2 by 8.6 and 3.9%, respectively. However, when the percentage of testing was 25% (0.25) in the uni-trait model, methods M3 and M4 outperformed methods M1 and M2 by 3.6 and 2.4%, respectively. Similar performance was observed in the other percentages of testing. 

We saw a clear superiority in the multi-trait model across data sets in terms of NRMSE in all percentages of testing (Figure 7, Table A3). In the uni-trait context, we observed that within testing 15% (0.15), M4 outperformed M1 and M2 by 8.0 and 5.8%, respectively, but M4 was worse than M3 by 1.7%. While within testing 25% (0.25), M4 outperformed M1 and M2 by 5.8 and 0.9%, respectively, and M4 was worse than M3 by 2.3%. Under the 50% (0.50) percentage of testing, M4 outperformed M1 and M2 by 3.9 and 0.6%, respectively, but M3 was better than M4 by 2.3%. Within testing 75% (0.75), M4 outperformed M1 and M2 by 2.6 and 7.5%, respectively, but M4 was worse than M3 by 0.3%. Finally, within testing 85% (0.85), M4 outperformed M1 and M2 by 0.6 and 4.9%, respectively, but M4 was worse than M3 by 1.6%. In the case of the uni-trait method, similar performance was observed among the four methods (Figure 7, Table A3). 

### 3.4. Assessing the Benefits of Sparse Testing Methods

In Table 1, we provide the comparison of two scenarios of breeding experiments: scenario 1 with 250 lines in each of the four environments (225 new lines + 25 checks) and 1000 plots available, and scenario 2 with 4500 lines in each of the four environments (4450 new lines + 50 checks) and 18,000 plots available. Each of these scenarios was compared with sparse designs with the following percentage of training data: 85, 75, 50, 25 and 15%. The standard is defined as the conventional breeding strategy, where all lines are evaluated in each environment.

Under scenario 1 we observed that the number of new lines for evaluation could be increased without a relevant increase in the budget, from 225 (Standard) to 269 (85% of training), 308 (75% of training), 475 (50% of training), 975 (25% of training) and 1624 (15% of training), which means increasing the new lines to be evaluated by 19.56% (85% training), 36.89% (75% of training), 111.11% (50% training), 333.33% (25% of training) and 628.78% (15% training). While we reach these increases without increasing the number of plots, instead of being replicated four times (one in each environment), each of the lines is now replicated 3.4 (85% training), 3 (75% training), 2 (50% training), 1 (25% training) and 0.6 (15% training) times, respectively. This means that the reduction in replication of lines is 15% (85% training), 25% (75% training), 50% (50% training), 75% (25% training) and 85% (15% training). Table 1 shows the comparison between the standard design and each percentage of sparse testing for other parameters (each row of Table 1 represents a different parameter).

For scenario two, we observed (Table 1) that new lines can be evaluated without a relevant increase in the budget, from 4450 (Standard) to 5244 (85% of training), 5950 (75% of training), 8950 (50% of training), 17,950 (25% of training) and 29,950 (15% of training), which implies increasing the new lines to be evaluated by 17.84% (85% training), 33.71% (75% of training), 101.12% (50% training), 303.37% (25% of training) and 573.03% (15% training). While we reach these increases with no increase in the number of plots, instead of replicating four times (one in each environment), each of the lines is now replicated 3.4 (85% training), 3 (75% training), 2 (50% training), 1 (25% training) and 0.6 (15% training) times. This indicates that the reduction in replication of lines is 15% (85% training), 25% (75% training), 50% (50% training), 75% (25% training) and 85% (15% training).

## 4. Discussion

Currently, GS methodology is being explored for its potential benefits, but its accuracy is influenced by many factors, making it difficult to optimize all of them simultaneously. As such, GS predictions are not yet accurate enough to be used routinely by plant and animal breeders.

In this vein, sparse testing methods are being studied to save significant resources in implementing GS methodology; however, it is still unclear which sparse testing methods are most efficient. The objective of this study was to better understand the efficiency of sparse testing with two large data sets and with two smaller data sets. These methods were implemented under a multi-trait and uni-trait framework to study their behavior in prediction accuracy. Additionally, we provided a cost-benefit analysis of implementing sparse testing methods.

As expected, we found that the best performance of the sparse testing methods was observed under a multi-trait model, and M3 and M4 were slightly better than sparse M1 and M2. However, we found that M3 was more consistent and robust, in addition to being efficient from a computational point of view. Although M4 and M3 were the best in terms of prediction performance, it is important to note that for larger training and testing sets, M4 was computationally inefficient, and since M3 displayed similar performance, it provides a better alternative to M4. However, when possible, M4 is the preferred option because the machinery of IBD and comparing treatments (genotypes) under more uniform conditions also reduces experimental error and increases precision. The primary factor in explaining the better performance of M3 and M4 compared to M1 and M2 is that M3 and M4 guarantee better connectivity between training and testing sets. However, M4 does not always outperform M3, as we observed that the larger the data set, the less difference there is in prediction performance between M3 and M4.

However, in the allocation of lines to environments, M4 is different from methods M1, M2 and M3 since M4 is based on the balanced incomplete block experimental design that uses a criterion of optimality to perform the allocation of lines to locations, thus potentially increasing efficiency. However, the nature of the data sets used for implementation plays an important role in the similar performance between M3 and M4 and among the four methods since the material (lines) of these data sets are homogeneous and possess a strong degree of relatedness. We also expect less differences in terms of prediction performance between methods M3 and M4 when the number of lines (treatments) increase because the numbers are large, making any randomization relatively good. The good performance of M3 is because it is also a type of incomplete block design but with not an optimal allocation as is done under an IBD experimental design providing less biased estimates.

It is important to be aware that for the successful implementation of the sparse testing methods evaluated in this research, the analyses of data in two stages is required because these sparse testing methods are applied in a second stage for evaluating the prediction performance of untested lines in tested environments which, as illustrated in this research, can save significant resources as only a subset of lines are evaluated in each location (environments), and the remaining lines are predicted. However, the second stage requires valid BLUEs (or BLUPs) that should be computed considering the experimental design in which the lines were evaluated in each location (environment). Note that since a two-stage process was performed, those lines allocated to a location (environment) can be evaluated in any experimental design, and after harvesting the traits of interest, this experimental design should be used for computing the BLUEs (BLUPs) of the lines evaluated in each location.

For this reason, when method M4 is used in the second stage, the implication is that a valid and efficient experimental design was already used in the first stage to estimate the appropriate BLUEs (or BLUPs) that consider the spatial variability of the field in which the cultivar was evaluated. In the second stage, we built the genomic training-testing with the aim of improving the prediction of the untested line in tested environments. However, in the second stage, when method M4 is used, the goal is to efficiently allocate the lines to locations to guarantee good connectivity between the lines in different locations, thus improving the prediction accuracy of the cultivar to be predicted.

However, under the cost-benefits analysis (see Table 1), we clearly observed the savings breeders could achieved using a sparse genomic testing approach. For example, in a sparse testing design with a training-testing scenario of 85–15%, under a fixed budget, we can increase the number of lines under evaluation by at least 17%. Under a sparse testing scenario of 50–50% for training-testing, the same fixed budget increased the lines under evaluation by at least 101.12%. Certainly, the larger the percentage of testing regarding the percentage of training, the larger the benefits of the sparse testing method; however, we do not expect these scenarios to be successful in all breeding programs. Nevertheless, in the four data sets evaluated, even in the scenario of training-testing of 15–85%, we observed a strong prediction performance, and the percentage of increase of new lines evaluated was at least 573%. While this scenario is not practical because it implies a fraction of replicates (0.6) of each line in the experiment, it does illustrate the benefits that can be obtained using a sparse testing methodology.

## 5. Conclusions

Using four data sets, we evaluated the prediction performance of four sparse testing methods (M1, M2, M3 and M4) under multi-trait and uni-trait models and under various scenarios of training-testing partitions. We found that the best accuracies were observed under a multi-trait model and the worst under a uni-trait model. We also observed that sparse testing methods M3 and M4 were slightly better than methods M1 and M2. Additionally, we found that the prediction accuracy, even in the more extreme scenario of training-testing (15–85%), is still competitive with the more relaxed scenario of training-testing (85–15%), which is of paramount importance since under this scenario the efficiency of the sparse testing methodology is very high. Even under a less extreme scenario of training-testing, 50–50% for training-testing, we increased the lines under evaluation by 101.12% with the same fixed budget, which helps to significantly increase the efficiency of the GS methodology. While these findings cannot be easily extrapolated for other data sets, they illustrate the great benefits that plant breeders can obtain from implementing sparse testing designs for genomic prediction.

## Figures and Tables

**Figure 1 genes-14-00927-f001:**
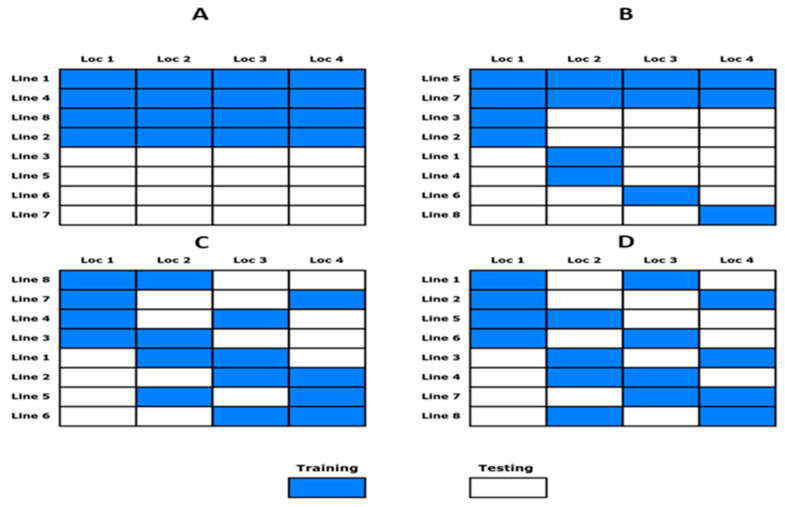
Allocation methods with eight lines and four locations for a partition, with 50% of lines as training and 50% of lines as testing. (**A**) M1 denotes the allocation of some lines in all locations, (**B**) M2 denotes the allocation of a subset of lines with some shared lines in locations, (**C**) M3 denotes the random allocation of some lines to locations under incomplete locations, and (**D**) M4 denotes the allocation of a fraction of lines to locations using the IBD method.

**Figure 2 genes-14-00927-f002:**
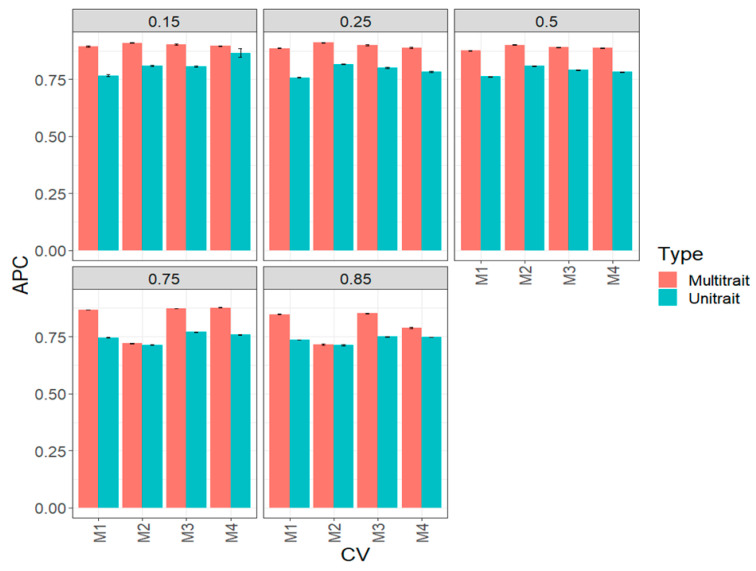
Prediction performance for the complete (big) maize data set in terms of average Pearson’s correlation (APC) of the four methods of sparse testing (M1, M2, M3 and M4) under unit-trait and multi-trait models for 5 percentages of testing: 15% (0.1), 25% (0.25), 50% (0.5), 75% (0.75) and 85% (0.85).

**Figure 3 genes-14-00927-f003:**
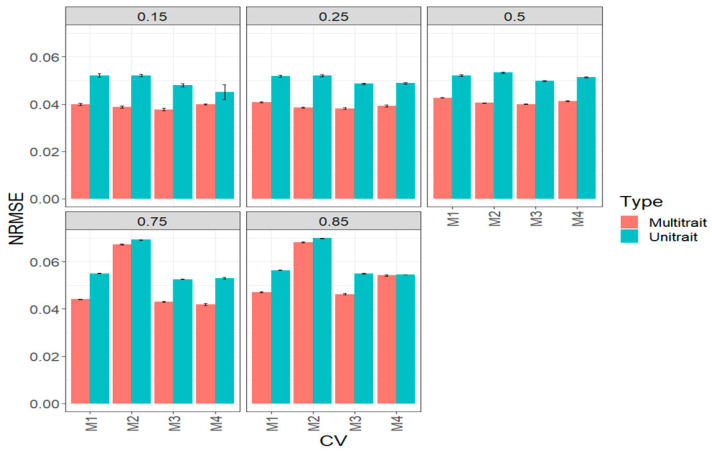
Prediction performance for the complete (big) maize data set in terms of normalized root mean square error (NRMSE) of the four methods of sparse testing (M1, M2, M3 and M4) under unit-trait and multi-trait models for 5 percentages of testing: 15% (0.1), 25% (0.25), 50% (0.5), 75% (0.75) and 85% (0.85).

**Figure 4 genes-14-00927-f004:**
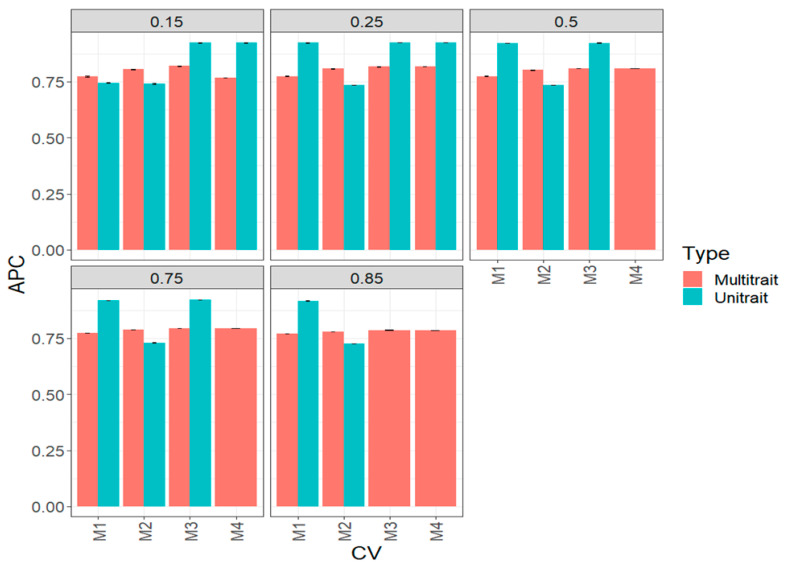
Prediction performance for the complete (big) wheat data set in terms of average Pearson’s correlation (APC) of the four methods of sparse testing (M1, M2, M3 and M4) under unit-trait and multi-trait models for 5 percentages of testing: 15% (0.1), 25% (0.25), 50% (0.5), 75% (0.75) and 85% (0.85).

**Figure 5 genes-14-00927-f005:**
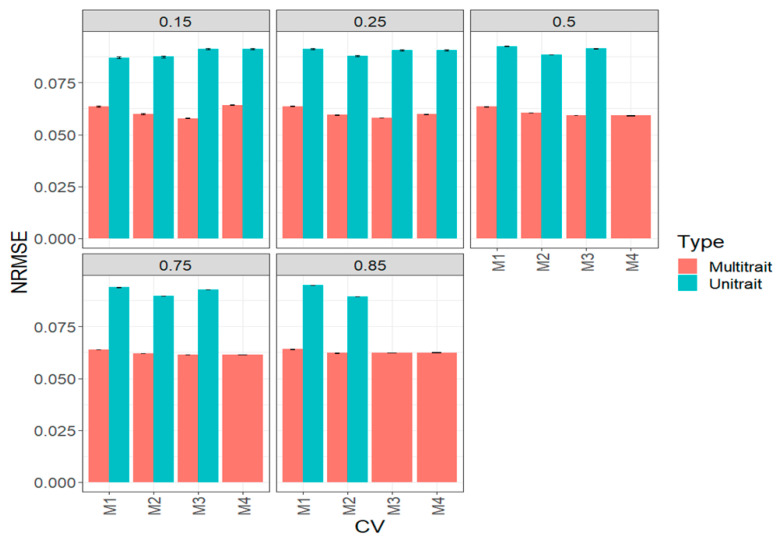
Prediction performance for the complete wheat data set in terms of normalized root mean square error (NRMSE) of the four methods of sparse testing (M1, M2, M3 and M4) under unit-trait and multi-trait models for 5 percentages of testing: 15% (0.1), 25% (0.25), 50% (0.5), 75% (0.75) and 85% (0.85).

**Figure 6 genes-14-00927-f006:**
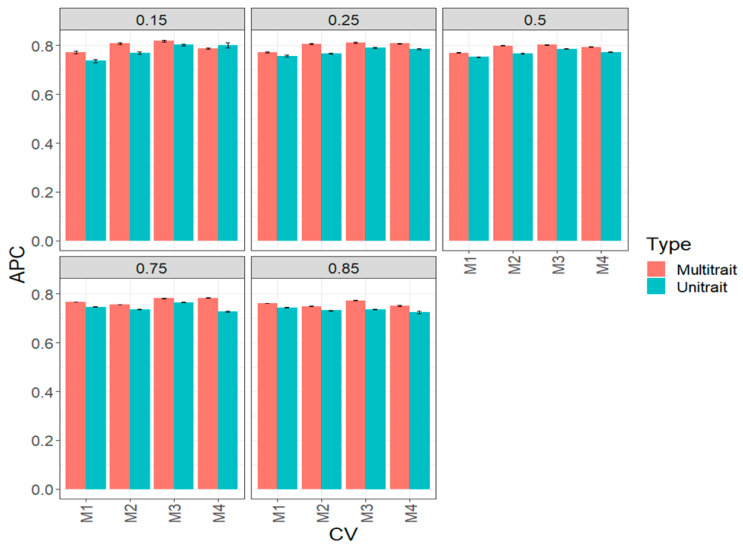
Prediction performance across data sets in terms of average Pearson’s correlation (APC) of the four methods of sparse testing (M1, M2, M3 and M4) under unit-trait and multi-trait models for 5 percentages of testing: 15% (0.1), 25% (0.25), 50% (0.5), 75% (0.75) and 85% (0.85).

**Figure 7 genes-14-00927-f007:**
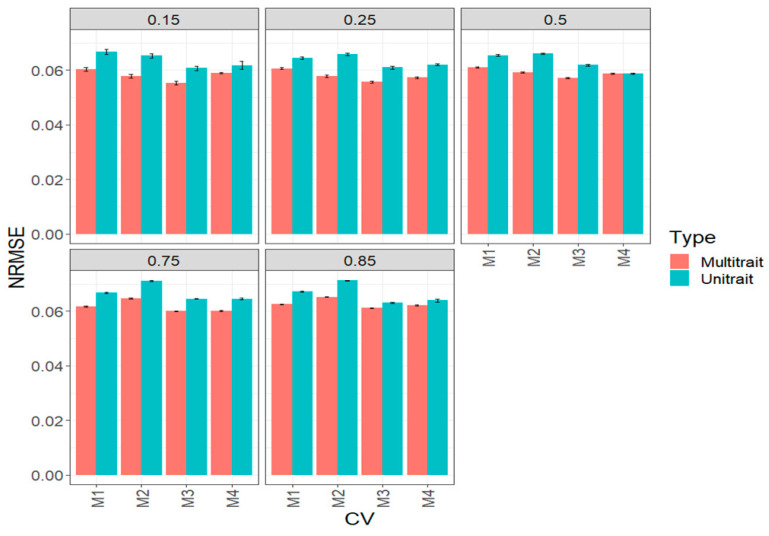
Prediction performance across data sets in terms of normalized root mean square error (NRMSE) of the four methods of sparse testing (M1, M2, M3 and M4) under unit-trait and multi-trait models for 5 percentages of testing: 15% (0.1), 25% (0.25), 50% (0.5), 75% (0.75) and 85% (0.85).

**Table 1 genes-14-00927-t001:** Gains or loss of using sparse methods (Incomplete designs) regarding conventional method (Standard) with different % of training sets (trn) using 4500 and 250 lines.

		Sparse Designs with Different % of trn Data					Gains (or Loss) of Sparse Designs for Each % of trn				
Concept	Standard	85	75	50	25	15	85	75	50	25	15
Scenario 1											
Total trts	250	294	333	500	1000	1667	17.60	33.20	100.00	300.00	566.80
New lines	225	269	308	475	975	1642	19.56	36.89	111.11	333.33	629.78
Checks	25	25	25	25	25	25	0.00	0.00	0.00	0.00	0.00
Reps	1	0.85	0.75	0.5	0.25	0.15	−15.00	−25.00	−50.00	−75.00	−85.00
Locs	4	4	4	4	4	4	0.00	0.00	0.00	0.00	0.00
R	4	3.4	3	2	1	0.6	−15.00	−25.00	−50.00	−75.00	−85.00
Total_plots	1000	1000	1000	1000	1000	1000	0.00	0.00	0.00	0.00	0.00
Total trts	250	294	333	500	1000	1667	17.60	33.20	100.00	300.00	566.80
Plots/trt	4.44	3.72	3.25	2.11	1.03	0.61	−16.36	−26.95	−52.63	−76.92	−86.30
Scenario 2											
Total trts	4500	5294	6000	9000	18000	30000	17.64	33.33	100.00	300.00	566.67
New lines	4450	5244	5950	8950	17950	29950	17.84	33.71	101.12	303.37	573.03
Checks	50	50	50	50	50	50	0.00	0.00	0.00	0.00	0.00
Reps	1	0.85	0.75	0.5	0.25	0.15	−15.00	−25.00	−50.00	−75.00	−85.00
Locs	4	4	4	4	4	4	0.00	0.00	0.00	0.00	0.00
R	4	3.4	3	2	1	0.6	−15.00	−25.00	−50.00	−75.00	−85.00
Tot_Plots	18000	18000	18000	18000	18000	18000	0.00	0.00	0.00	0.00	0.00
Total trts	4500	5294	6000	9000	18000	30000	17.64	33.33	100.00	300.00	566.67
Plots/trt	4.04	3.43	3.03	2.01	1.00	0.60	−15.14	−25.21	−50.28	−75.21	−85.14

## Data Availability

The phenotypic and genomic maize and wheat data employed in this study for the complete (Big) data and for the small data comprising only 250 lines can be downloaded from the following link https://hdl.handle.net/11529/10548813.

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
