# Peer review of "Optimizing Sparse Testing for Genomic Prediction of Plant Breeding Crops"

_genes, 2023, doi:10.3390/genes14040927_

Round 1

Reviewer 1 Report

Introduction section was well written, enough and clear.

The method were described well.

Why 15% and 25% training set proportion were used not defined. There test ratio was too low. (Line 274)

Why didn’t use R2 for comparison.

In Table 1, 2 and 3. It is too hard to understand and compare. Please use different tables for Multi and Uni traits.

Results can be given more clearly.

Conclusion wasn’t enough. So many results but no real conclusion.

Author Response

RESPONSE TO REVIEWER 1

Open Review

Quality of English Language

( ) English very difficult to understand/incomprehensible
( ) Extensive editing of English language and style required
( ) Moderate English changes required
(x) English language and style are fine/minor spell check required
( ) I am not qualified to assess the quality of English in this paper

Yes

Can be improved

Must be improved

Not applicable

Does the introduction provide sufficient background and include all relevant references?

( )

(x)

( )

( )

Are all the cited references relevant to the research?

(x)

( )

( )

( )

Is the research design appropriate?

( )

(x)

( )

( )

Are the methods adequately described?

( )

(x)

( )

( )

Are the results clearly presented?

( )

( )

(x)

( )

Are the conclusions supported by the results?

( )

( )

(x)

( )

Comments and Suggestions for Authors

Introduction section was well written, enough and clear.

The method were described well.

Why 15% and 25% training set proportion were used not defined. There test ratio was too low. (Line 274)

RESPONSE: See new version of the paper. See lines 284-286.

Why didn’t use R2 for comparison.

RESPONSE: Many thanks. We did not use R2 because correlation and MSE are the most popular metrics used in the context of genomic prediction.

In Table 1, 2 and 3. It is too hard to understand and compare. Please use different tables for Multi and Uni traits.

RESPONSE: Thanks for the good advice. We have move Tables 1-3 to the Appendix because the same information can be observed from the figures. Furthermore, the tables have a column to distinguish results from multi-trait and uni-trait; for this reason, we believe that it is better for the readers to have all information for Multi-trait and uni-trait in 3 tables rather than in 6 tables

Results can be given more clearly.

RESPONSE: We agree, thanks. By taking the large tables from the main text and leave only the figures we think we have significantly clarified the results.

Conclusion wasn’t enough. So many results but no real conclusion.

RESPONSE: Thanks for your concern. In the conclusion we have highlighted the following statement that expresses the most important conclusion from this study…. We found the best accuracies were observed under the multi-trait model and the worst under a uni-trait model. We also observed that sparse testing methods M3 and M4 were slightly better than methods M1 and M2. Additionally, we found that the prediction accuracy even in the more extreme scenario of training-testing (15-85%) is still competitive with the more relaxed scenario of training-testing (85-15%), which is of paramount importance since under this scenario the efficiency of the sparse testing methodology is very high.

Submission Date

01 March 2023

Date of this review

12 Mar 2023 18:31:38

Reviewer 2 Report

Dear authors! Thank you for submitting the manuscript. The paper submitted for review is devoted to methods for sparse testing allocation of lines to environments under multi-environmental trails for genomic prediction of unobserved lines. The work is interesting and understandable for a very narrow circle of specialists. For its greater attractiveness, I recommend in the Introduction and in the Conclusion to formulate the main tasks and results of this work in a more understandable language for a wide range of readers. I consider it wrong to use the Pearson criterion in this work. This kniterium applies to the normal distribution of data. However, in biology, as a rule, primary data have a non-normal distribution, so criteria such as Pearson's test are not suitable for testing validity. Explain why you applied this particular criterion. Names of species in Latin in the article are not always written in italics. Write in the Conclusion and in the Introduction how promising is your work in the field of identifying the most productive strains of microorganisms - producers in biotechnology? In Conclusion, you write: “We found the best accuracies were observed under the multi-trait model and the worst under a uni-trait model.” I don't see anything new in this conclusion. This is quite logical as it is.

Respectfully Yours, Reviewer

March 13, 2023

Author Response

RESPONSE TO REVIEWER 2

Top of Form

Open Review

Quality of English Language

( ) English very difficult to understand/incomprehensible
( ) Extensive editing of English language and style required
(x) Moderate English changes required
( ) English language and style are fine/minor spell check required
( ) I am not qualified to assess the quality of English in this paper

Yes

Can be improved

Must be improved

Not applicable

Does the introduction provide sufficient background and include all relevant references?

( )

( )

(x)

( )

Are all the cited references relevant to the research?

( )

( )

(x)

( )

Is the research design appropriate?

( )

(x)

( )

( )

Are the methods adequately described?

(x)

( )

( )

( )

Are the results clearly presented?

(x)

( )

( )

( )

Are the conclusions supported by the results?

( )

(x)

( )

( )

Comments and Suggestions for Authors

Dear authors! Thank you for submitting the manuscript. The paper submitted for review is devoted to methods for sparse testing allocation of lines to environments under multi-environmental trails for genomic prediction of unobserved lines. The work is interesting and understandable for a very narrow circle of specialists. For its greater attractiveness, I recommend in the Introduction and in the Conclusion to formulate the main tasks and results of this work in a more understandable language for a wide range of readers.

RESPONSE: Thanks for your comments. We rewritten the last paragraph of the introduction to be clearer with the objectives of our study. See lines 108-120 of the revised manuscript.

I consider it wrong to use the Pearson criterion in this work. This kniterium applies to the normal distribution of data. However, in biology, as a rule, primary data have a non-normal distribution, so criteria such as Pearson's test are not suitable for testing validity. Explain why you applied this particular criterion.

RESPONSE: The main reason we used Pearson´s correlation as metric for evaluating the prediction accuracy is because this is very popular in the context of genomic prediction. However, due to the issues you mentioned we also provide the normalized root mean square error (NRMSE). However, it is interesting to point out that our findings agree in this case using both metrics for this reason we believe that even that the Pearson´s correlation is not always a good option for measuring prediction accuracy in this case the results provided are in agreement with those of the NRMSE. For this reason, we believe that both metrics complement each other and give us a better picture of the prediction performance of the four evaluated sparse methods.

Names of species in Latin in the article are not always written in italics.

 Write in the Conclusion and in the Introduction how promising is your work in the field of identifying the most productive strains of microorganisms - producers in biotechnology?

RESPONSE: In the Introduction we have written the following statement at lines 107-121…. This study aimed to optimize allocation methods to improve genomic prediction accuracy of sparse testing by evaluating four genomic sparse testing strategies for allocating cultivars to environments. This study addressed four objectives that have not been investigated in any previous studies: (1) to determine if there are differences in prediction ability between the four genomic sparse testing allocation methods; (2) to study if there are significant differences between the four strategies of sparse testing under a uni-trait (UT) and multi-trait prediction framework; (3) to evaluate the performance of the four sparse testing strategies with large and small trials; and (4) to quantify the various benefits of implementing this genomic sparse testing allocation of lines to environments strategies. To achieve these objectives, two real data sets from CIMMYT were used—one maize and one wheat—with one data set containing over 450 lines and the other over 4500 lines. To assess performance with small trial sizes from each of these two data sets, a random sample of 250 lines in each environment was obtained, and the four sparse testing methods were evaluated using this resulting data set.

In Conclusion, you write: “We found the best accuracies were observed under the multi-trait model and the worst under a uni-trait model.” I don't see anything new in this conclusion. This is quite logical as it is.

RESPONSE: See lines 484-491 in the Conclusion….. We found the best accuracies were observed under the multi-trait model and the worst under a uni-trait model. We also observed that sparse testing methods M3 and M4 were slightly better than methods M1 and M2. Additionally, we found that the prediction accuracy even in the more extreme scenario of training-testing (15-85%) is still competitive with the more relaxed scenario of training-testing (85-15%), which is of paramount importance since under this scenario the efficiency of the sparse testing methodology is very high

Respectfully Yours, Reviewer

RESPONSE: Many thanks for your time invested in revising this article

March 13, 2023

Submission Date

01 March 2023

Date of this review

13 Mar 2023 08:00:28

Bottom of Form

© 1996-2023 MDPI (Basel, Switzerland) unless otherwise stated

Round 2

Reviewer 1 Report

Suggested revisions were done or clarified.

Author Response

RESPONSE TO REVIEWER 2 EDITOR

Dear Authors,

Thank you for submitting your revised paper to our scientific journal. We are pleased to inform you that the two independent reviewers who evaluated your submission are satisfied with the changes you have made, and we have decided to accept your paper for publication pending the completion of a minor request regarding script availability.

We really appreciate that you have provided access to your dataset, which is an essential aspect of reproducible science. However, to further facilitate the ability of researchers to build on your work and to promote greater transparency in the research process, we would like to request that you release the script code used in your study.

Specifically, we suggest that you provide the code necessary to run the methods M1, M2, M3, and M4 on your dataset and to conduct cross-validation assessments. This will enable other researchers to replicate and build on your findings, increasing the impact and visibility of your research.

We recommend that you deposit the code in a recognized, public repository such as GitHub, upload it as supplementary information to your publication or add it to the CIMMYT webpage containing your dataset.

Thank you for considering our journal for your research, and we look forward to receiving your revised manuscript.

Dear Editor:

Thanks for your letter and for the opportunity to publish in GENES. Yes, indeed we agree with you in relation to the concept that results  from a scientific publication should be totally reproducible.. Therefore, data and computer codes must be provided to readers.

For this article we have shared the data and now in another Supplemental Materials we provided the R codes for fitting the 4 models (M1-M4) outlined in the article and some required information to run these R codes.

We hope that with this addition the article could be published with no further delay..
